# Do Sex Differences in Respiratory Burst Enzyme Activities Exist in Human Immunodeficiency Virus-1 Infection?

**DOI:** 10.3390/medsci4040019

**Published:** 2016-11-15

**Authors:** Mathias Abiodun Emokpae, Beatrice Aghogho Mrakpor

**Affiliations:** Department of Medical Laboratory Science, School of Basic Medical Sciences, College of Medical Sciences, University of Benin, Benin 300001, Nigeria; aghoghomrakpor@gmail.com

**Keywords:** human immunodeficiency virus type 1, respiratory burst enzymes, sex differences

## Abstract

Studies have shown that human immunodeficiency virus type 1 (HIV-1) disproportionally affects more females than males. Affected individuals are susceptible to infections due to depressed immunity, qualitative defects in phagocytic function and altered phagocytosis as well as lowered oxidative burst capacity. This study seeks to determine whether sex differences exist in serum activities of respiratory burst enzymes in HIV-1–infected female and male subjects. Serum myeloperoxidase, catalase and superoxide dismutase activities were assayed in 170 confirmed HIV-1 positive and 50 HIV-1 negative subjects using ELISA. Data were analyzed using Student’s *t*-test and *p* values of less than 0.05 were considered significant. The measured enzyme activities were significantly higher (*p* < 0.001) in females than males in HIV-1 negative subjects while no sex differences were observed in HIV-1 positive subjects. The absence of sex differences in the activities of respiratory burst enzymes in HIV-1 infection may be due to immune activation as a result of active phagocytic leukocytes, immune reactivity and inflammation.

## 1. Introduction

Human immunodeficiency virus type 1 (HIV-1) infection is a global health challenge that disproportionally affects more females than males [1]. Studies have shown that about 60% of all subjects with HIV-1 infection are women, especially in sub-Saharan Africa [2]. Apart from social and political factors, biological and genetic dimorphisms have been suggested to be responsible for the differences in disease courses and outcome [3,4]. Following seroconversion after HIV-1 infection, females have 40% less viral load and more cluster of differentiation 4 (CD4)^+^ cells than males [3], but at the same level of viremia, disease severity and progression to acquired immune disease are faster in women [5,6,7,8]. Some of the reasons attributed to the sex disparity in HIV-1 infection are increased levels of generalized immune activation [9,10,11] and elevated inflammatory activity [12,13]. One of the observed disparities in sex differences was innate immunity [9,12]. The activity of enzymes in the body can be changed either by their rate of synthesis and secretion from the organ of origin, the distribution in the extracellular compartments or the rate and routes of elimination as well as inactivation [14,15]. These factors may be influenced by individual variability, gender, diseases, drugs or physical activities [2,9,12]. These factors are often considered to ensure the appropriate and meaningful use of enzymes in clinical diagnosis and management of patients [14,15].

Respiratory burst enzymes are contributory factors to HIV-1 disease progression [16,17] and are induced by the production of reactive oxygen species (ROS). The deficiency of total antioxidant status might also increase oxidation stress, adversely affecting the immune response and predisposing the patient to drug toxicity [18]. Myeloperoxidase (MPO) is an enzyme present in polymorphonuclear leukocytes which performs a vital role in the destruction of phagocytosed microorganisms. MPO activity is directly associated with the activation of phagocytic leukocytes due to invading pathogens which often lead to the destruction of the leukocytes [19]. Respiratory burst refers to the sudden increase in oxygen consumption by reactive oxidants, resulting in the production of hypochlorus acid (HOCl) when cells are invaded by pathogens [20,21]. When pathogens are ingested by neutrophils or monocytes, their oxygen consumption increases suddenly. The respiratory burst enzyme (myeloperoxidase) converts hydrogen peroxide (H_2_O_2_) to HOCl in the presence of chloride. The formed HOCl helps to eliminate phagocytosed pathogens [22]. Studies have indicated that increased susceptibility of HIV-1 infected subjects to bacterial infection may be due to phagocyte dysfunction [23,24]. It has been reported that qualitative defects in phagocytic function in HIV-1 infection can lead to impaired chemotaxis, altered phagocytosis, lowered oxidative burst capacity and altered bacterial killing [25,26], which may exhibit a sex disparity. This is an area of interest for this study which was designed to determine if sex differences exist in the serum activities of respiratory burst enzymes and whether these differences also exist or not in HIV-1 infection. 

## 2. Patients and Methods

### 2.1. Selection of Study Participants

The study participants were consecutively enrolled and comprised of 220 subjects that consisted of 120 confirmed HIV-1 positive individuals receiving highly active antiretroviral therapy (HAART) (20 males with mean age of 34.9 ± 0.6 years and 100 females with mean age of 33.5 ± 0.5 years), 50 newly diagnosed HAART-naive HIV-1 positive subjects (24 males with mean age of 32.3 ± 0.4 years and 26 females with mean age of 32.1 ± 0.2 years) and 50 HIV-1 negative (apparently healthy) individuals recruited from among staff and students of Federal Medical Centre, Yenagoa (controls, 25 males with mean age of 32.6 ± 0.3 years and 25 females with mean age of 31.9 ± 0.4 years).

### 2.2. Inclusion and Exclusion Criteria

All the confirmed HIV-1 subjects attending the antiretroviral therapy (ART) clinics at the Federal Medical Center, Yenagoa, that gave consent were included in the study. All HIV-1 seronegative individuals who had an illness or infection (chest infections, bacterial endocarditis) or smoke cigarettes that may affect respiratory burst enzymes as well as those who did not give consent were excluded from the study.

### 2.3. Ethical Consideration

The study protocol was reviewed and approved by the Ethics and Research Committee of the Federal Medical Center (ethical code FMC/EC/01/2016, dated 21^st^ January, 2016), Yenagoa, Bayelsa State, before the commencement of the study. Informed consent was sought and obtained from all participants and utmost confidentiality of information was maintained.

### 2.4. Specimen Collection and Analytical Methods

Six milliliters of blood sample were collected by venous puncture; 3 mL were dispensed into a plain tube, while the remaining 3 mL were emptied into bottles containing ethylene diamine tetra-acetic acid (EDTA). The sample in the plain container was allowed to clot at room temperature. The clotted sample was centrifuged at 3000 rpm for 10 min. The serum was separated into plain containers and stored at −20 °C before it was analysed. The serum superoxide dismutase (SOD), catalase (CAT) and MPO were assayed by Enzyme Linked Immunosorbent Assay (ELISA) technique using reagent kits (HS-901811) supplied by Wkea Medical supplies Corp.,Changchun, China. The experiments were done according to the manufacturer’s protocol.

The CD4^+^ count was estimated using Fluorescence Activated Cell Sorter (Facs Flow Cytometer) count system, Lincolnshire, IL, USA. 

### 2.5. Statistical Analysis

The data generated from this study were analyzed by the statistical software SPSS version IBM 21 (SPSS Inc., Chicago, IL, USA) for Windows. Categorical variables were compared using Student’s *t*-test. A *p* value < 0.05 was considered statistically significant.

## 3. Results

Table 1 shows the comparison of respiratory burst enzyme activities in HIV-1 negative male and female subjects. The activities of the measured enzymes were significantly higher (*p* < 0.001) in females.

Table 2 shows the comparison of respiratory burst enzymes in HIV-1 positive male and female subjects. The activity levels were not statistically significant (*p* > 0.05) between males and females.

Table 3 shows the comparison of respiratory burst enzyme activities in HIV-1 positive male and female subjects on HAART. The levels show no statistically significant difference (*p* > 0.05) between sexes.

## 4. Discussion

This study was conducted to determine whether sex differences exist in the activities of respiratory burst enzymes in HIV-1 negative subjects and to know if sex disparity is affected by HIV-1 infection. HIV-1 infected subjects are immunocompromised, and prone to opportunistic infections and leukocyte activation as a result of opsonophagocytosis which is actively taking place [27,28]. Sex differences in the activities of respiratory burst enzymes were observed in HIV-1 negative control subjects with activities of MPO (16.1%), CAT (19.3%) and SOD (11.3%) being higher in women than in men. The observed sex differences are consistent with previous studies [29,30,31,32]. These authors also reported that MPO and SOD expression and activity were higher in women than in men. The gender differences in the MPO activity were attributed to higher levels of neutrophils, a cell type rich in MPO. Males were reported to have a lower systemic neutrophil count [29] and lower neutrophil survival than females [30]. Saraymen et al. [33] evaluated the activities of polymorphonuclear leukocyte enzymes in healthy subjects and observed that only SOD activity was significantly higher in females than males, while no sex-dependent correlation between CAT and MPO activities was observed in healthy polymorphonuclear leukocytes, which suggested that CAT and MPO activities were not sex-dependent [33]. The sex differences could also be due to the modulating influence of female sex hormones [3,4,30]. Leukocytes use MPO to generate oxidants (ROS) that can initiate lipid peroxidation and oxidation of low density lipoprotein cholesterol (atherogenic form), which are recognized by macrophage scavenger receptors [34]. It was observed that the effects of MPO-generated HOCl provoked a biphasic response in human endothelial cells (EC) [35]. Whereas low levels of HOCl (<10 µmol/L) result in EC activation and the generation of tissue factor mRNA, protein and tissue factor pathway activation, higher doses (30–50 µmol/L) provoke EC death by apoptotic mechanisms in which there is rapid caspase 3 activation, a decrease in EC B-cell lymphoma 2 (Bcl-2), cytochrome c release and DNA laddering [34]. In HIV-1 infected subjects, however, there were no sex differences in the measured respiratory burst enzymes. Sex-dependent disparities in the activities of respiratory burst enzymes have been reported in several disease conditions and in drug metabolism, both in human and animal studies [35,36,37,38,39,40,41]. Sex differences in the measured enzyme activities were reported in spontaneously hypertensive rats with SOD and CAT significantly higher in males than females. No sex differences were reported in borderline hypertensives and nine-week-old hypertensive Wistar-Kyoto rats [35].

The implication of sex differences in the expression of drug metabolizing and transport genes and disease susceptibility in a sex-dependent fashion has been of interest in health studies [36]. Sex differences in renal angiotensin converting enzyme 2 activity have been reported. Plasma renin activity was observed to be 27% higher in men than women in normotensive subjects [37]. Sex differences in biology may be due to sex hormone differences that occur during puberty, adulthood and menopause. They could also be due to differences in sex chromosome dosage (2X versus 1X and 0Y versus 1Y) [37]. Other authors using fibroblasts have reported that human X-linked genes escape X inactivation fully or partially with only few subsets of these genes that are highly expressed in females [38,39]. Parental imprinting was also suggested to be responsible for sex differences in biology. It may be involved in the potential difference in the effects of XX and XY sex chromosome complements [40]. Whereas the X chromosome in males is only imprinted maternally, the X chromosome in females is imprinted both maternally and paternally. Parental imprinting may impact gene function and bring about pathophysiological consequences [37,41].

The observed sex differences in the activities of respiratory burst enzymes in HIV-1–negative individuals were eliminated in HIV-1 positive subjects both on HAART and not on HAART. Even though women had lower viral loads and higher CD4^+^ cell counts during primary infection, they were reported to have a 1.6-fold higher risk of HIV-1 disease progression which ultimately equals the rate observed in men. We hypothesized that the elimination of the sex differences in the respiratory burst enzyme activities may be a contributing factor to disease progression in women. Immune activation has been recognized as a strong predictor for HIV-1 disease progression, which is independent of viral load [2]. Sex disparities in immunological responses to HIV-1 infection may be mediated through genetic variation and the effects of sex hormones [2,42,43,44], differences in anatomy and sex chromosome-associated factors [1]. The fewer symptoms and delayed diagnoses reported in women [45] may be associated with increased activities of respiratory burst enzymes before primary HIV-1 infection. The increase in respiratory burst enzyme activities could be an attempt to combat rapid viral replication in females. The respiratory burst leads to increased oxygen consumption and increased free radical generation which, in turn, leads to an increment of the inflammatory and apoptotic processes. The leakage of toxic substances generated from the activities of respiratory burst enzymes has been reported to damage the nearby cells [19]. In addition, the oxidants produced are toxic, immunosuppressive and have mutagenic effects [46,47]; these factors could contribute to the disease course and outcomes. Sex differences in the Toll-like receptor (TLR) response to HIV infection have been reported. It was suggested that the sex differences in TLR response may largely be responsible for HIV-1 disease progression irrespective of the viral load [2] and CD4^+^ cell counts. The secretion of interferon-α from plasmacytoid dendritic cells via TLR was reported to be significantly higher in women than men [48]. HIV-1 infected females had higher CD4^+^ cell counts than male subjects on HAART and not on HAART. This observation is consistent with previous studies [49,50]. Multivariate linear regression that demonstrated an independent association with sex of both HIV-1 RNA levels and CD4^+^ percentages was reported [49]. CD4^+^ cell counts were also observed to be higher in adult women with and without HIV-1 infection [50]. It was suggested that intrinsic biological differences in immunologic response to HIV-1 exist between males and females [49]. Our observation has important implications in understanding the biology of the antiviral immune response and possible management strategies.

## 5. Conclusions

The activities of respiratory burst enzymes, which were higher in HIV-1 negative female subjects, were significantly decreased and to equal those of the male counterparts in HIV-1 infection independent of the CD4^+^ cell counts. The absence of sex differences in the activities of respiratory burst enzymes may be a reflection of higher immune activation, immune reactivity, inflammation and subsequent differential disease course and outcome in HIV-1 infection in women compared to men.

## Figures and Tables

**Table 1 medsci-04-00019-t001:** Comparison of respiratory burst enzyme activities between male and female HIV-1 negative subjects.

Measured Parameters	Female *n* = 25	Male *n* = 25	*p* Value
CD4^+^ cell count (IU/mL)	805.1 ± 9.2	774.6 ± 6.1	*p* < 0.005
CAT (IU/L)	12.19 ± 0.06	10.97 ± 0.07	*p* < 0.001
SOD (ng/mL)	1.87 ± 0.03	1.68 ± 0.04	*p* < 0.001
MPO (ng/mL)	10.10 ± 0.28	8.70 ± 0.23	*p* < 0.001

CD4^+^: cluster of differentiation 4; CAT: catalase, SOD: superoxide dismutase, MPO: myeloperoxidase.

**Table 2 medsci-04-00019-t002:** Comparison of respiratory burst enzymes activities between male and female HIV-1 positive highly active antiretroviral therapy (HAART)-naïve subjects.

Measured Parameters	Female *n* = 26	Male *n* = 24	*p* Value
CD4^+^ cell count (IU/mL)	347.4 ± 17.7	245.7 ± 9.9	*p* < 0.001
CAT (IU/L)	15.44 ± 0.97	16.74 ± 1.92	*p* = 0.10
SOD (ng/mL)	1.56 ± 0.022	1.58 ± 0.045	*p* = 0.50
MPO (ng/mL)	8.06 ± 1.01	7.11 ± 1.72	*p* = 0.50

**Table 3 medsci-04-00019-t003:** Comparison of respiratory burst enzymes activities between male and female HIV-1 positive subjects on HAART.

Parameters	Female *n* = 100	Male *n* = 20	*p* Value
CD4^+^ cell count (IU/mL)	510.5 ± 12.7	376 ± 17.3	*p* < 0.001
CAT (IU/L)	18.24 ± 1.15	18.44 ± 2.82	*p* = 0.80
SOD (ng/mL)	1.49 ± 0.02	1.45 ± 0.04	*p* = 0.50
MPO (ng/mL)	9.67 ± 1.34	9.50 ± 2.82	*p* = 0.88

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
