# Peer review of "Do Sex Differences in Respiratory Burst Enzyme Activities Exist in Human Immunodeficiency Virus-1 Infection?"

_medsci, 2016, doi:10.3390/medsci4040019_

Reviewer 1 Report

Summary: The work presented is a clinical study that showed sex differences in the activity of respiratory burst enzymes in HIV infected as well as un-infected subjects.

Major comments:

1.     Tables 2 and 3 have a lot more female subjects in the comparisons. I’m wondering how this might have affected the statistics. Results from tables 2 and 3 are then compared to the results of table 1, HIV negative subjects which had an equal number of men and women. I am pointing this out because the author’s conclusions differed from reference 18, who had more subjects in their study (55 males and 54 females). The authors used reference 19, Sugiyama et al, as a support for their results. However, this paper used human cell-lines and is not a clinical study as authors claimed, and made no mention of sex differences as the authors claimed. If I am mistaken, the authors should provide the exact tables and figures from reference 19 that supported their conclusions. Please cite previous work appropriately. Thus, I am not convinced that the authors conclusively proved sex differences in activity of respiratory burst enzymes in HIV infected and uninfected subjects. The authors could randomly sample equal number of male and female subject results to see if unequal sample size is affecting their conclusions.

2.     The authors attribute difference in disease progression in females to the difference in the activity of respiratory burst enzymes in presence of HIV infection. But if there is no statistical difference in the respiratory burst enzyme activity in HIV infected men and women, then the disease progression should be similar in both groups, not different. I’d like the authors to discuss this.

3.     The paper can use good English editing. There are quite a few grammatical errors, and a good editing would make it easier to read.

4.     The paper does not quality as a “Research Article” in my opinion, due to the lack of depth in their investigation. This would be best classified as a brief report or brief communication.

Minor comments:

1.     More citations are needed in the introduction: lines 36-38, lines 46-50. Introduction should end with the goal of the study, which the authors instead did in lines 43-45.

2.     More details should be provided for study participates: age range, number of male and female subjects. Section 2.1 starts with “subjects who met the selection criteria”, before explaining what the criteria were. The number of female and male subjects was later clarified in tables 1-3 but should be mentioned earlier as well.

3.     Please specify the type of HIV that was focused on: HIV-1 or HIV-2.

4.     Typo: HoCL should be HOCL.

5.     The title of the paper seems to have random capitalization of words. It is also not very informative.

6.     Please check the reference list to make sure there are no spelling mistakes. The first author in reference 18 is Saraymen, not Suraymen, a mistake repeated in the manuscript text as well.

7.     Please keep spacing uniform throughout the manuscript: one space between words.

8.     Tables 1-3: CAT concentration is listed as (micro/Liter). Micro is just a prefix. Please provide proper units.

9.     Please provide more details of the ELISA method, and if a kit was used. Providing the name of the reagent supplier alone is not helpful in reproducing the results.

Author Response

Summary: The work presented is a clinical study that showed sex differences in the activity of respiratory burst enzymes in HIV infected as well as un-infected subjects.

Major comments:

1.    Tables 2 and 3 have a lot more female subjects in the comparisons. I’m wondering how this might have affected the statistics. Results from tables 2 and 3 are then compared to the results of table 1, HIV negative subjects which had an equal number of men and women. I am pointing this out because the author’s conclusions differed from reference 18, who had more subjects in their study (55 males and 54 females). The authors used reference 19, Sugiyama et al, as a support for their results. However, this paper used human cell-lines and is not a clinical study as authors claimed, and made no mention of sex differences as the authors claimed. If I am mistaken, the authors should provide the exact tables and figures from reference 19 that supported their conclusions. Please cite previous work appropriately. Thus, I am not convinced that the authors conclusively proved sex differences in activity of respiratory burst enzymes in HIV infected and uninfected subjects. The authors could randomly sample equal number of male and female subject results to see if unequal sample size is affecting their conclusions.

Response: Table 2 has 26 females and 24 males, this was a mix-up and highly regretted. You will notice that 50 subjects were enrolled as HIV infected subjects who were HAART naïve. Only table 3 has more females than males and the results still remained similar to that of table 2. Randomly sampled equal number of male and female subjects did not affect unequal sample size and conclusions made. Our observation has been duly compared with previous studies. See references 20, 21, 22,23 and 37.

2.    The authors attribute difference in disease progression in females to the difference in the activity of respiratory burst enzymes in presence of HIV infection. But if there is no statistical difference in the respiratory burst enzyme activity in HIV infected men and women, then the disease progression should be similar in both groups, not different. I’d like the authors to discuss this.

Response: It was suggested that intrinsic biologic differences in immunologic response to HIV exist between males and females [37]. Since infected women display increased levels of generalized immune-activation and experience the effect of elevated inflammatory activities than men.

3.    The paper can use good English editing. There are quite a few grammatical errors, and a good editing would make it easier to read.

Response: English editing and spelling mistakes have been corrected.

4.     The paper does not quality as a “Research Article” in my opinion, due to the lack of depth in their investigation. This would be best classified as a brief report or brief communication.

Minor comments:

1.    More citations are needed in the introduction: lines 36-38, lines 46-50. Introduction should end with the goal of the study, which the authors instead did in lines 43-45.

Response: More citations given; references 3 and 4. The aim of the study moved to the end of introduction section.

2.    More details should be provided for study participates: age range, number of male and female subjects. Section 2.1 starts with “subjects who met the selection criteria”, before explaining what the criteria were. The number of female and male subjects was later clarified in tables 1-3 but should be mentioned earlier as well.

Response: Demographic information included in materials and methods section.

3.    Please specify the type of HIV that was focused on: HIV-1 or HIV-2.

Response: HIV-1 specified.

4.     Typo: HoCL should be HOCL Response: Corrected.

5.     The title of the paper seems to have random capitalization of words. It is also not very informative.

Response: Title changed to: Sex Differences in Respiratory Burst Enzymes Activities may be eliminated by Human Immunodeficiency virus infection.

6.     Please check the reference list to make sure there are no spelling mistakes. The first author in reference 18 is Saraymen, not Suraymen, a mistake repeated in the manuscript text as well. Response: Duly corrected

7.     Please keep spacing uniform throughout the manuscript: one space between words.

8.     Tables 1-3: CAT concentration is listed as (micro/Liter). Micro is just a prefix. Please provide proper units. Response: Unit of measurement was U/L

9.     Please provide more details of the ELISA method, and if a kit was used. Providing the name of the reagent supplier alone is not helpful in reproducing the results.

Response: General principle of ELISA given. 

Reviewer 2 Report

Well written paper.

My few comments: 

Would useful to motivate the paper is the overall take home message and the implication of the findings in HIV management. I think the paper lacked this. ie what is the implication of the findings?

Selection criteria not well described.  The paper would read better if the screening process is well described for both HIV positive and HIV negative participants. 

Blood sample assays? were the tests performed for everyone? 

I think the main idea is immune response. It would be very helpful and informative to present the results by the CD4 levels.

Present the exact p values. > 0.05 is OK but could be more informative if exact p values are provided.

Author Response

Comments and Suggestions for Authors

Well written paper.

My few comments: 

Would useful to motivate the paper is the overall take home message and the implication of the findings in HIV management. I think the paper lacked this. ie what is the implication of the findings?

Response: Implication provided thus: Our observation has important implications for understanding the biology of antiviral immune response and possibly management strategies. 

Selection criteria not well described.  The paper would read better if the screening process is well described for both HIV positive and HIV negative participants. Response: This section has been adjusted as suggested. 

Blood sample assays? were the tests performed for everyone? Response: Tests were performed for all participants.

I think the main idea is immune response. It would be very helpful and informative to present the results by the CD4 levels. Response: CD4 levels included in tables 1-3.

Present the exact p values. > 0.05 is OK but could be more informative if exact p values are provided.

Response :p-values are presented as required by the journal.

Note: All corrections and additional information are highlighted in red ink in the text.

Round  2

Reviewer 1 Report

I am disappointed to see that the authors chose to ignore a few of my comments and only made cosmetic changes to their manuscript when a major revision was required. I will briefly outline my issues with this revised work:

Introduction is still very sparsely cited. I had specifically mentioned two sets of line numbers in my first review that needed citations, and the authors added only two references for the first set. The second set of line numbers remain as they were in the original version.

The same grammatical errors in the original version are still there, even though the authors said they had fixed those. I understand it is difficult for authors whose first language is not english (I am in this group too), but one should seek out colleagues with better command over the language to help out in such matters.

The simplest thing that authors could have done was to specify which HIV type they were talking about: HIV-1 or HIV-2. Yet, they only sporadically corrected this at couple of places, leaving the ambiguous term "HIV" scattered throughout their manuscript.

The author's response to my 2nd major commend in the first review is not satisfactory.

The title is still is not very informative, nor is it significantly different from the previous one.

For CAT, the authors list units of measurement as U/L, but what is U?

Everyone knows the principle of ELISA (still, it was good to provide details about the method). My intent was to get the authors to say if they used any kit, the manufacturer/catalog number of the kit; this is so common I am surprised I have to ask for this twice. The goal is to give enough information so that your results can be reproduced. Which monoclonal antibody was used? Which HRP polyclonal antibody was used? What was the wash buffer? These are basic details that everyone mentions, unless an ELISA kit is used. If a kit is used then the manufacturer name is mentioned.

Given the change in male and female sample sizess, I'd implore the authors to double check all their numbers for any other major mix-ups.

This work does not contribute substantial material, as is a requirement for publication under the type "Article". This best fits under a brief communication. The authors did not respond to this suggestion.

Author Response

 Comments and Suggestions for Authors

I am disappointed to see that the authors chose to ignore a few of my comments and only made cosmetic changes to their manuscript when a major revision was required. I will briefly outline my issues with this revised work:

Introduction is still very sparsely cited. I had specifically mentioned two sets of line numbers in my first review that needed citations, and the authors added only two references for the first set. The second set of line numbers remain as they were in the original version.

Response: More citations added. See references 9,10,11,12,12,14 and 15.

The same grammatical errors in the original version are still there, even though the authors said they had fixed those. I understand it is difficult for authors whose first language is not english (I am in this group too), but one should seek out colleagues with better command over the language to help out in such matters.

Response: We have checked and corrected grammatical errors.

The simplest thing that authors could have done was to specify which HIV type they were talking about: HIV-1 or HIV-2. Yet, they only sporadically corrected this at couple of places, leaving the ambiguous term "HIV" scattered throughout their manuscript.

Response: HIV type 1 has been specified anywhere HIV is used

The author's response to my 2nd major commend in the first review is not satisfactory.

Response: The following statements were added to explain lack of sex differences of respiratory burst enzymes in subjects with HIV-1 infection which exists only in non-infected subjects; Even though women had lower viral load and higher CD4 cell counts during primary infection, they were reported to have a 1.6-fold higher risk of HIV-1 disease progression which ultimately equals the rate observed in men. We hypothesized that the elimination of the sex differences in the Respiratory burst enzymes activities may be a contributing to the disease progression in women. Since immune activation has been recognized as a strong predictor for HIV-1 disease progression which is independent of viral load [2]. Sex disparity in immunological responses to HIV-1 infection may be mediated through genetic variation and the effects of sex hormones [2,42-44], differences in anatomy and sex chromosome-associated factors [1]. The fewer symptoms and delay diagnosis reported in women [45] may be associated with increased activities of respiratory burst enzymes before primary infection in females.

The title is still is not very informative, nor is it significantly different from the previous one.

Response: New title is:  Do Sex Differences in Respiratory Burst Enzymes Activities Exist in Human Immunodeficiency virus-1 infection?

For CAT, the authors list units of measurement as U/L, but what is U?

Response: IU/L is international unit per litre.

Everyone knows the principle of ELISA (still, it was good to provide details about the method). My intent was to get the authors to say if they used any kit, the manufacturer/catalog number of the kit; this is so common I am surprised I have to ask for this twice. The goal is to give enough information so that your results can be reproduced. Which monoclonal antibody was used? Which HRP polyclonal antibody was used? What was the wash buffer? These are basic details that everyone mentions, unless an ELISA kit is used. If a kit is used then the manufacturer name is mentioned.

Response: ELISA kits were used.

I am flummoxed by the drastic change in male and female sample sizes from 34 and 136 to 24 and 26 in the revised version after I pointed out the inequality in sample sizes. Given that this is such a major change, I'd implore the authors to double check all their numbers for any other major mix-ups. Response: It has been checked, we are sorry for the mix-ups.

This work does not contribute substantial material, as is a requirement for publication under the type "Article". This best fits under a brief communication. The authors did not respond to this suggestion.

Response: We believed that this manuscript qualifies for publication under the type original article because it does not only contributes to knowledge but it contains the various components of original article, ie hypothesis, background, materials and methods, results, interpretation of findings and discussion. 

Reviewer 2 Report

Most of my comments have been addressed. Would have been helpful to present actual p values rather than presenting it as >0.05.

Author Response

Reviewer 2

Most of my comments have been addressed. Would have been helpful to present actual p values rather than presenting it as >0.05.

Response: The actual p-values have been presented for those with p>0.05.

Table 3: Comparison of respiratory burst enzymes activities between male and female HIV positive subjects on HAART.

PARAMETERS

Male

Female  

P-VALUE  

No.   of subjects

CD4 cell counts(IU/mL)

n = 20

376±17.3

n = 100

510.5±12.7

P<0.001< span="">

CAT(IU/L)

18.44±2.82

18.24±1.15

P=0.80

SOD   (ng/mL)

1.45±0.04

1.49±0.02

P=0.50

MPO   (ng/mL)

9.50±2.82

9.67±1.34

P=0.88

Round  3

Reviewer 1 Report

I disagree with the author's assessment that this work qualifies as an 'original article' because it has all components of it such as hypothesis, background, methods, and so forth. The entire point of this work is that the activity of respiratory burst enzymes shows gender disparity in HIV-1 negative subjects compared to HIV-1 positive subjects. They do not do any follow-up in vitro studies to answer WHY they see this gender disparity. Thus, this work does not further our understanding on how HIV-1 infection progresses in men vs women, it only finds that there are gender differences. Do other viral infectious also show this gender difference? Or is this effect specific to HIV-1 infection? This investigation is thus very shallow and did not seek to investigate the reasons for the trend they observed. In my opinion, the contribution is minimal and suited for publication as a brief communication. Even brief communications have to detail the hypothesis, methods, and interpretation of the results.

While I appreciate the authors' effort to expand the literature review, the authors can be more careful in addressing other concerns: there are still places where the subtype (HIV-1) is still not mentioned (see Results section).

 The ELISA method details are unchanged from the last time: I still don't see catalog numbers from Wkea Medical Supplies, whose kits they used OR any specific details about the enzymes/buffers/antibodies. Please specify the catalog number in order to provide enough details so that an independent lab can reproduce the results.